# ROBUST REINFORCEMENT LEARNING VIA ADVERSARIAL TRAINING WITH LANGEVIN DYNAMICS

## ABSTRACT

We re-think the Two-Player Reinforcement Learning (RL) as an instance of a distribution sampling problem in infinite dimensions. Using the powerful Stochastic Gradient Langevin Dynamics, we propose a new two-player RL algorithm, which is a sampling variant of the two-player policy gradient method. Our new algorithm consistently outperforms existing baselines, in terms of generalization across differing training and testing conditions, on several MuJoCo environments.

## 1 INTRODUCTION

Reinforcement learning (RL) promise automated solutions to many real-world tasks with beyond-human performance. Indeed, recent advances in policy gradient methods (Sutton et al., 2000; Silver et al., 2014; Schulman et al., 2015; 2017) and deep reinforcement learning have demonstrated impressive performance in games (Mnih et al., 2015; Silver et al., 2017), continuous control (Lillicrap et al., 2015), and robotics (Levine et al., 2016) towards this grand challenge.

Despite the success of deep RL, the progress is still upset by the fragility in real-life deployments. In particular, majority of these methods fail to perform well when there is some difference between training and testing scenarios, thereby posting serious safety and security concerns. To this end, learning policies that are *robust* to environmental shifts, mismatched configurations, and even mismatched control actions is becoming increasingly more important.

A powerful framework to learning robust policies is to interpret the changing of the environment as an adversarial perturbation. This notion naturally lends itself to a two-player minimax problem involving a pair of agents, a protagonist and an adversary, where the protagonist learns to fulfill the original task goals while being robust to the disruptions generated by its adversary.

Two prominent examples along this research vein, differing in how they model the adversary, are the Robust Adversarial Reinforcement Learning (RARL) (Pinto et al., 2017) and Noisy Robust Markov Decision Process (NR-MDP) (Tessler et al., 2019). Despite achieving impressive performance in practice, these existing frameworks heavily rely on heuristic algorithms, and hence, suffer from lack of theoretical guarantees, even in idealistic cases with infinite data as well as computational power.

One critical challenge in robust RL setting is that while maximizing rewards is a well-studied subject in classical/deep RL, the needed two-player minimax version is significantly more complicated to solve both in theory and practice. For instance, Tessler et al. (2019) prove that it is in fact strictly suboptimal to directly apply (deterministic) policy gradient steps to their NR-MDP max-min objectives. Owing to the lack of a better algorithm, the policy gradient is nonetheless still employed in their experiments; similar comments also apply to (Pinto et al., 2017).

Our paper precisely bridges this gap between theory and practice in previous works, by proposing the first theoretically convergent algorithm for robust RL. Our key idea is to switch from optimizing the max-min reward to *sampling* from the optimal *randomized policies*, which corresponds to finding a mixed Nash Equilibrium (NE) (Nash et al., 1950) in the max-min objective.

It is a classical fact in game theory that, while deterministic minimax objective is often ill-posed, the mixed NE is well-behaved under very mild assumptions. Furthermore, algorithmic approaches to finding mixed NE with finite strategies have been studied extensively (Freund & Schapire, 1999; Nemirovski, 2004) and is recently extended to the case of infinite strategies (Hsieh et al., 2019). In particular, (Hsieh et al., 2019) show that, by using the Stochastic Gradient Langevin Dynamics

(SGLD) (Welling & Teh, 2011) to take samples from randomized strategies, one can find a mixed NE of Generative Adversarial Networks (Goodfellow et al., 2014).

Our work introduces the same mixed NE perspective to the max-min objectives in robust RL, and substantiate the ensuing theoretical framework with extensive experimental evidence. We apply the new sampling framework to the well-known Deep Deterministic Policy Gradient (DDPG) method in the scope of NR-MDP. We demonstrate that the new algorithm achieves clearly superior performance in its generalization capabilities.

Intriguingly, we also observe that the idea of mixed strategy in single-player RL (i.e., the non-robust or one-player formulation) can lead to substantially more robust policies over the standard DDPG algorithm. More precisely, we represent the agent's policy as a distribution over deterministic policies, and aim to sample from the distribution $\mu(\pi) \propto \exp(-\frac{1}{\sigma}J(\pi))$ where $\pi$ denotes a deterministic policy, $J$ the associated expected reward, and $\sigma$ a temperature parameter going to $0$ during training.

Our numerical evidence demonstrates that DDPG combined with this sampling approach for the actor update leads to learned policies that generalize better to unseen MDPs, when compared to the state-of-the-art DDPG variant of Tessler et al. (2019) while using similar computational resources.

## 2 BACKGROUND

### 2.1 STOCHASTIC GRADIENT LANGEVIN DYNAMICS (SGLD)

For any probability distribution $p(z) \propto \exp(-g(z))$, the Stochastic Gradient Langevin Dynamics (SGLD) Welling & Teh (2011) iterates as

$$z_{k+1} \; \leftarrow \; z_k - \gamma \left[ \widehat{\nabla_z g(z)} \right]_{z=z_k} + \sqrt{2\gamma}\epsilon\xi_k, \tag{1}$$

where $\gamma$ is the step-size, $\widehat{\nabla_z g(z)}$ is an unbiased estimator of $\nabla_z g(z)$, $\epsilon > 0$ is a temperature parameter, and $\xi_k \sim \mathcal{N}(0, I)$ is a standard normal vector, independently drawn across different iterations. In some cases, the convergence rate of SGLD can be improved by scaling the noise using a positive-definite symmetric matrix $C$. We thus define a preconditioned variant of the above update equation 1 as follows:

$$z_{k+1} \; \leftarrow \; z_k - \gamma C^{-1} \left[ \widehat{\nabla_z g(z)} \right]_{z=z_k} + \sqrt{2\gamma}\epsilon C^{-\frac{1}{2}}\xi_k. \tag{2}$$

In the experiments, we use a RMSProp-preconditioned version of the SGLD (Li et al., 2016).

### 2.2 INFINITE-DIMENSIONAL BI-LINEAR GAMES

In this section, we review some of the key results from (Hsieh et al., 2019). We denote the set of all probability measures on $\mathcal{Z}$ by $\mathcal{P}(\mathcal{Z})$, and the set of all functions on $\mathcal{Z}$ by $\mathcal{F}(\mathcal{Z})$. Given a (sufficiently regular) function $h : \Theta \times \Omega \to \mathbb{R}$, consider the following objective (a two-player game with *infinitely* many strategies):

$$\max_{p \in \mathcal{P}(\Theta)} \min_{q \in \mathcal{P}(\Omega)} f(p, q) \; := \; \mathbb{E}_{\theta \sim p} \left[ \mathbb{E}_{\omega \sim q} [h(\theta, \omega)] \right]. \tag{3}$$

A pair $(p^*, q^*)$ achieving the max-min value in equation 3 is called a *mixed Nash Equilibrium* (NE). Define the operator $G : \mathcal{P}(\Omega) \to \mathcal{F}(\Theta)$, and its adjoint operator $G^\dagger : \mathcal{P}(\Theta) \to \mathcal{F}(\Omega)$ as follows:

$$Gq(\theta) \; := \; \mathbb{E}_{\omega \sim q}[h(\theta, \omega)] \in \mathcal{F}(\Theta)$$

$$G^\dagger p(\omega) \; := \; \mathbb{E}_{\theta \sim p}[h(\theta, \omega)] \in \mathcal{F}(\Omega).$$

Denoting $\langle p, g \rangle := \mathbb{E}_{z \sim p}[g(z)]$ for any probability measure $p$ and function $g$ on $\mathcal{Z}$, we can write $f$ as $f(p, q) = \langle p, Gq \rangle = \langle G^\dagger p, q \rangle$. Furthermore, the derivative (the analogue of gradient in infinite dimension) of $f(p, q)$ with respect to $p$ is simply $Gq$, and the derivative of $f(p, q)$ with respect to $q$ is $G^\dagger p$; i.e., $\nabla_p f(p, q) = Gq$, and $\nabla_q f(p, q) = G^\dagger p$.

---

**Algorithm 1** Infinite-Dimensional Entropic Mirror Descent-v2

---

**Input:** Initial distributions $p_1, q_1$, and learning rate $\eta$
**for** $t = 1, 2, \ldots, T-1$ **do**
$$p_{t+1}(\theta) \propto \exp\left(+\eta \sum_{s \leq t} Gq_s(\theta)\right)$$
$$q_{t+1}(\omega) \propto \exp\left(-\eta \sum_{s \leq t} G^\dagger p_s(\omega)\right)$$
**end for**
**Output:** $\bar{p}_T = \frac{1}{T} \sum_{t=1}^{T} p_t$ and $\bar{q}_T = \frac{1}{T} \sum_{t=1}^{T} q_t$

---

**Algorithm 2** Approximate Infinite-Dimensional Entropic Mirror Descent

---

**Input:** $\omega_1, \theta_1 \leftarrow$ random initialization, SGLD step-size $\{\gamma_t\}_{t=1}^{T}$, thermal noise of SGLD $\{\epsilon_t\}_{t=1}^{T}$, warmup steps for SGLD $\{K_t\}_{t=1}^{T}$, exponential damping factor $\beta$, standard normal noise $\xi_k, \xi_k'$.
**for** $t = 1, 2, \ldots, T-1$ **do**
$\quad \bar{\omega}_t, \omega_t^{(1)} \leftarrow \omega_t$ ; $\bar{\theta}_t, \theta_t^{(1)} \leftarrow \theta_t$
$\quad$ **for** $k = 1, 2, \ldots, K_t$ **do**
$$\theta_t^{(k+1)} \leftarrow \theta_t^{(k)} + \gamma_t \left[\nabla_\theta \widehat{h(\theta, \omega_t)}\right]_{\theta = \theta_t^{(k)}} + \sqrt{2\gamma_t}\epsilon_t \xi_k$$
$$\omega_t^{(k+1)} \leftarrow \omega_t^{(k)} - \gamma_t \left[\nabla_\omega \widehat{h(\theta_t, \omega)}\right]_{\omega = \omega_t^{(k)}} + \sqrt{2\gamma_t}\epsilon_t \xi_k'$$
$$\bar{\omega}_t \leftarrow (1-\beta)\bar{\omega}_t + \beta\omega_t^{(k+1)} \ ; \ \bar{\theta}_t \leftarrow (1-\beta)\bar{\theta}_t + \beta\theta_t^{(k+1)}$$
$\quad$ **end for**
$\quad \omega_{t+1} \leftarrow (1-\beta)\omega_t + \beta\bar{\omega}_t$ ; $\theta_{t+1} \leftarrow (1-\beta)\theta_t + \beta\bar{\theta}_t$
**end for**
**Output:** $\omega_T, \theta_T$.

---

Conceptually, problem (3) can be solved via the so-called infinite-dimensional entropic mirror descent; see Algorithm 1. Hsieh et al. (2019) have proved the convergence (to the mixed NE) rate of Algorithm 1. But this algorithm is infinite-dimensional and requires infinite computational power to implement. For practical interest, by leveraging the SGLD sampling techniques and using some practical relaxations, Hsieh et al. (2019) have proposed a simplified variant of Algorithm 1. The pseudocode for their resulting algorithm can be found in Algorithm 2.

## 3 TWO-PLAYER MARKOV GAMES

**Markov Decision Process:** We consider a Markov Decision Process (MDP) represented by $\mathcal{M}_1 := (\mathcal{S}, \mathcal{A}, T_1, \gamma, P_0, R_1)$, where the state and action spaces are denoted by $\mathcal{S}$ and $\mathcal{A}$ respectively. We focus on continuous control tasks, where the actions are real-valued, *i.e.*, $\mathcal{A} = \mathbb{R}^d$. $T_1 : \mathcal{S} \times \mathcal{S} \times \mathcal{A} \to [0, 1]$ captures the state transition dynamics, *i.e.*, $T_1(s' \mid s, a)$ denotes the probability of landing in state $s'$ by taking action $a$ from state $s$. Here $\gamma$ is the discounting factor, $P_0 : \mathcal{S} \to [0, 1]$ is the initial distribution over states $\mathcal{S}$, and $R_1 : \mathcal{S} \times \mathcal{A} \to \mathbb{R}$ is the reward.

**Two-Player Zero-Sum Markov Games:** Consider a two-player zero-sum Markov game Littman (1994); Perolat et al. (2015), where at each step of the game, both players simultaneously choose an action. The reward each player gets after one step depends on the state and the joint action of both players. Furthermore, the transition kernel of the game is controlled jointly by both the players. In this work, we only consider simultaneous games, not the turn-based games.

This game can be described by an MDP $\mathcal{M}_2 = (\mathcal{S}, \mathcal{A}, \mathcal{A}', T_2, \gamma, R_2, P_0)$, where $\mathcal{A}$ and $\mathcal{A}'$ are the continuous set of actions the players can take, $T_2 : \mathcal{S} \times \mathcal{A} \times \mathcal{A}' \times \mathcal{S} \to \mathbb{R}$ is the state transition probability, and $R_2 : \mathcal{S} \times \mathcal{A} \times \mathcal{A}' \to \mathbb{R}$ is the reward for both players. Consider an agent executing a policy $\mu : \mathcal{S} \to \mathcal{A}$, and an adversary executing a policy $\nu : \mathcal{S} \to \mathcal{A}'$ in the environment $\mathcal{M}$. At each timestep $t$, both players observe the state $s_t$ and take actions $a_t = \mu(s_t)$ and $a_t' = \nu(s_t)$. In the zero-sum game, the agent gets a reward $r_t = R_2(s_t, a_t, a_t')$ while the adversary gets a negative reward $-r_t$.

This two-player zero-sum Markov game formulation has been used to model the following robust RL settings:

- Robust Adversarial Reinforcement Learning (RARL) (Pinto et al., 2017), where the power of the adversary is limited by the action space $\mathcal{A}'$ of the adversary.
- Noisy Robust Markov Decision Process (NR-MDP) (Tessler et al., 2019), where $\mathcal{A}' = \mathcal{A}$, $T_2\left(s_{t+1} \mid s_t, a_t, a_t'\right) = T_1\left(s_{t+1} \mid s_t, \bar{a}_t\right)$, and $R_2\left(s_t, a_t, a_t'\right) = R_1\left(s_t, \bar{a}_t\right)$, with $\bar{a}_t = (1 - \delta)a_t + \delta a_t'$, for $\delta \in (0, 1)$. The power of the adversary is limited by $\delta$.

In our adversarial game, we consider the following performance objective:

$$J\left(\mu, \nu\right) \;=\; \mathbb{E}\left[\sum_{t=1}^{\infty} \gamma^{t-1} r_t \;\middle|\; \mu, \nu, \mathcal{M}\right],$$

where $\sum_{t=1}^{\infty} \gamma^{t-1} r_t$ be the random cumulative return. In particular, we consider the parameterized policies $\{\mu_\theta : \theta \in \Theta\}$, and $\{\nu_\omega : \omega \in \Omega\}$. By an abuse of notation, we denote $J\left(\theta, \omega\right) = J\left(\mu_\theta, \nu_\omega\right)$. We consider the following objective:

$$\max_{\theta \in \Theta} \min_{\omega \in \Omega} \; J\left(\theta, \omega\right). \tag{4}$$

Note that $J$ is non-convex/concave in both $\theta$ and $\omega$. Instead of solving equation 4 directly, we focus on the mixed strategy formulation of equation 4. In other words, we consider the set of all probability distributions over $\Theta$ and $\Omega$, and we search for the optimal distribution that solves the following program:

$$\max_{p \in \mathcal{P}(\Theta)} \min_{q \in \mathcal{P}(\Omega)} \; f\left(p, q\right) \;:=\; \mathbb{E}_{\theta \sim p}\left[\mathbb{E}_{\omega \sim q}\left[J\left(\theta, \omega\right)\right]\right]. \tag{5}$$

Then, we can use the techniques from Section 2.2 to solve the above problem.

## 4 Experiments

In this section, we demonstrate the effectiveness of using infinite-dimensional sampling techniques to solve the robust RL problem.

**Two-Player DDPG:** As a case study, we consider NR-MDP setting with $\delta = 0.1$ (as recommended in Section 6.3 of (Tessler et al., 2019)). We design a two-player variant of DDPG (Lillicrap et al., 2015) algorithm by adapting the Algorithm 2. As opposed to standard DDPG, in two-player DDPG two actor networks output two deterministic policies, the protagonist and adversary policies, denoted by $\mu_\theta$ and $\nu_\omega$. The critic is trained to estimate the Q-function of the joint-policy. The gradients of the protagonist and adversary parameters are given in Proposition 5 of (Tessler et al., 2019). The resulting algorithm is given in Algorithm 3.

We compare the performance of our algorithm against the baseline algorithm proposed in (Tessler et al., 2019) (see Algorithm 4). (Tessler et al., 2019) have suggested a training ratio of $1 : 1$ for actors and critic updates. Note that the action noise is injected while collecting transitions for the replay buffer. In Fujimoto et al. (2018), authors noted that the action noise drawn from the Ornstein-Uhlenbeck Uhlenbeck & Ornstein (1930) process offered no performance benefits. Thus we also consider uncorrelated Gaussian noise.

**Setup:** We evaluate the performance of Algorithm 3 and Algorithm 4 on standard continuous control benchmarks available on OpenAI Gym Brockman et al. (2016) utilizing the MuJoCo environment Todorov et al. (2012). Specifically, we benchmark on eight tasks: Walker, Hopper, Half-Cheetah, Ant, Swimmer, Reacher, Humanoid, and InvertedPendulum. Details of these environments can be found in Brockman et al. (2016) and on the GitHub website.

The Algorithm 3 implementation is based on the codebase from (Tessler et al., 2019). For all the algorithms, we use a two-layer feedforward neural network structure of (64, 64, tanh) for both actors (agent and adversary) and critic. The optimizer we use to update the critic is Adam Kingma & Ba (2015) with a learning rate of $10^{-3}$. The target networks are soft-updated with $\tau = 0.999$.

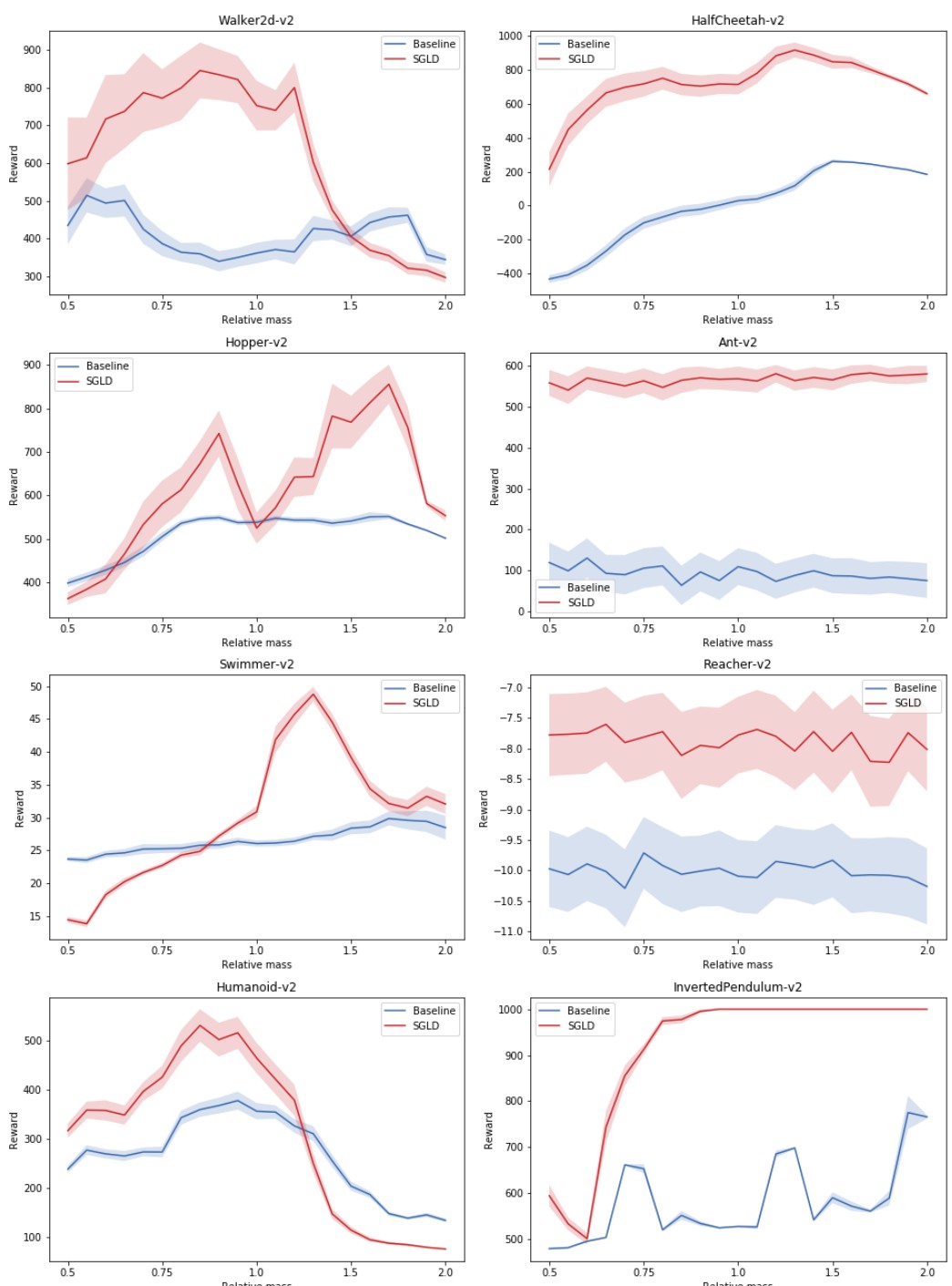

Figure 1: Average performance (over 5 seeds) of Algorithm 3 (—), and Algorithm 4 (—), under the NR-MDP setting with $\delta = 0.1$. The evaluation is performed without adversarial perturbations, on a range of mass values not encountered during training.

For the baseline, the actors are trained with RMSProp optimizer. For our algorithm, the actors are updated according to Algorithm 2 with warmup steps $K_t = \min\left\{15, \lfloor(1 + 10^{-5})^t\rfloor\right\}$, and thermal noise $\sigma_t = \sigma_0 \times (1 - 5 \times 10^{-5})^t$. The hyperparameters that are not related to exploration (see Table 1) are identical to both algorithms that are compared.

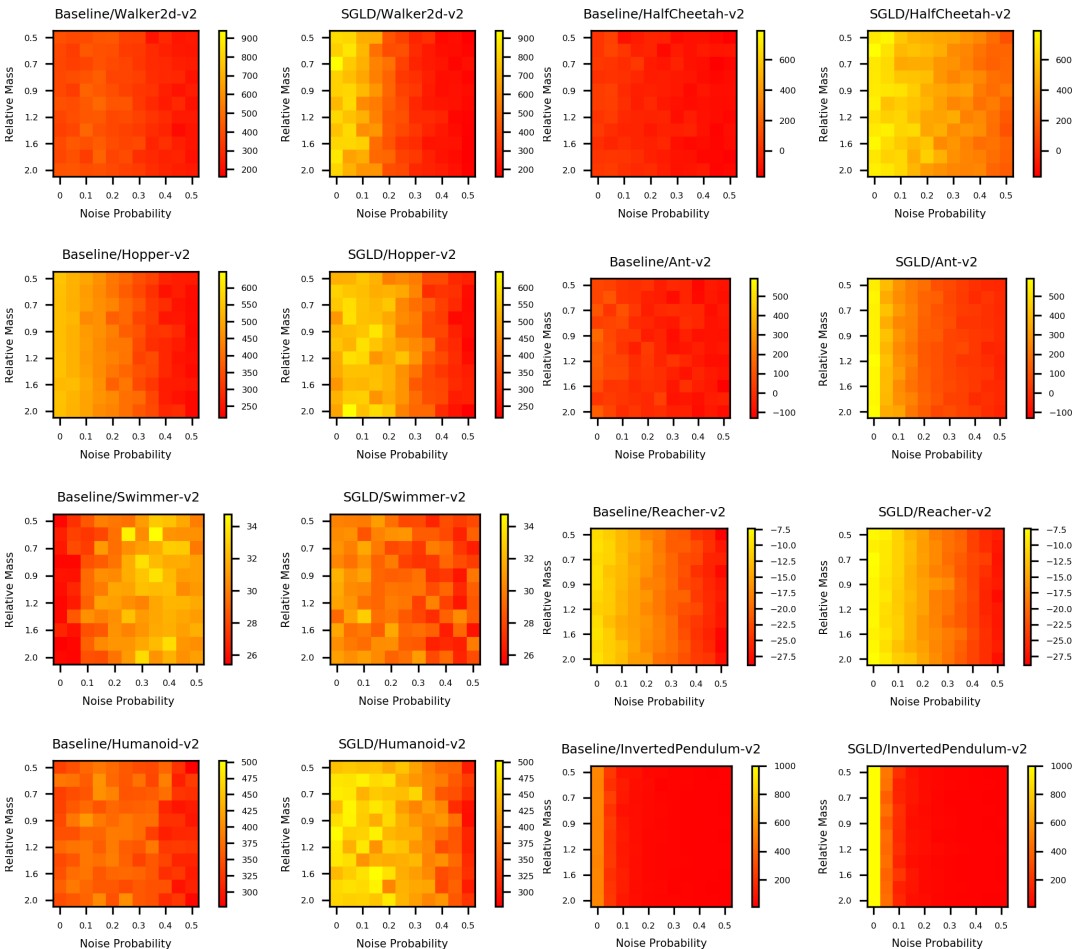

Figure 2: Average performance (over 5 seeds) of Algorithm 3 (—), and Algorithm 4 (—), under the NR-MDP setting with $\delta = 0.1$. The evaluation is performed on a range of noise probability and mass values not encountered during training.

And we tuned only the exploration-related hyper-parameters (for both algorithms) by grid search: (a) Algorithm 3 with $(\sigma_0, \sigma) \in \{10^{-2}, 10^{-3}, 10^{-4}, 10^{-5}\} \times \{0, 0.01, 0.1, 0.2, 0.3, 0.4\}$ ; (b) Algorithm 4 with $\sigma \in \{0, 0.01, 0.1, 0.2, 0.3, 0.4\}$. For each algorithm-environment pair, we identified the best performing exploration hyperparameter configuration (see Tables 2 and 3).

Each algorithm is trained on 0.5M samples (i.e., 0.5M time steps in the environment). We run our experiments, for each environment, with 5 different seeds. The exploration noise is turned off for evaluation.

**Evaluation:** We evaluate the robustness of both algorithms under different testing conditions, and in the presence of adversarial disturbances in the testing environment. We train both algorithms with the standard mass variables in OpenAI Gym. At test time, we evaluate the learned policies by changing the mass values (without adversarial perturbations) and estimating the cumulative rewards. As shown in Figure 1, our Algorithm 3 outperforms the baseline Algorithm 4 in terms of robustness. We also evaluate the robustness of the learned policies under both test condition changes, and adversarial disturbances (see Figure 2).

**One-Player DDPG:** We evaluate the robustness of one-player variants of Algorithm 3, and Algorithm 4, i.e., we consider the NR-MDP setting with $\delta = 0$. In this case, we set $K_t = 1$ for

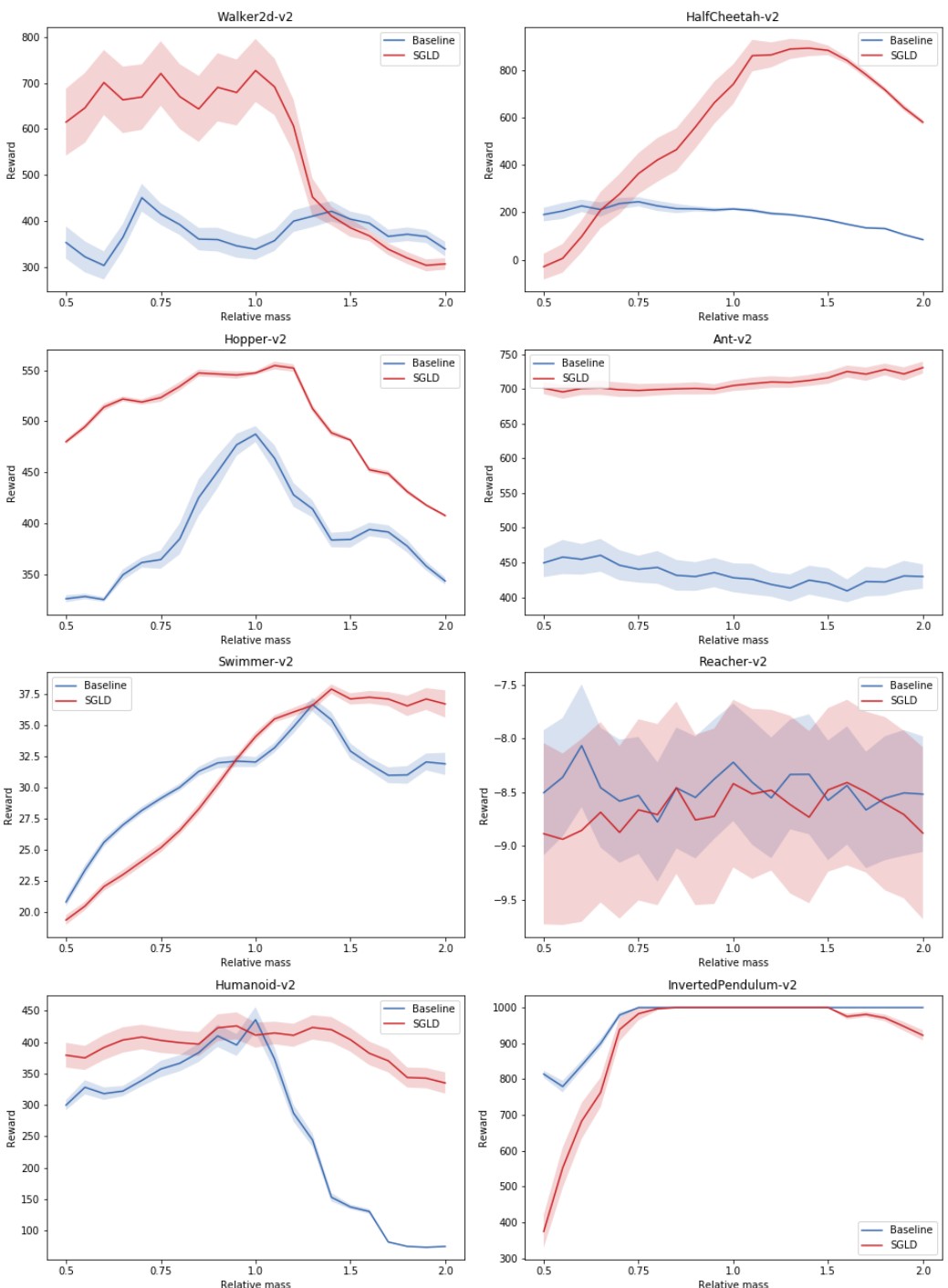

Figure 3: Average performance (over 5 seeds) of Algorithm 3 (—), and Algorithm 4 (—), under the NR-MDP setting with $\delta = 0$. The evaluation is performed without adversarial perturbations, on a range of mass values not encountered during training.

Algorithm 3 (this choice of $K_t$ makes the computational complexity of both algorithms equal). The results are presented in Figures 3 and 4.

Here, we remark that Algorithm 3 with $\delta = 0$, and $K_t = 1$ is simply the standard DDPG with actor being updated by preconditioned version of SGLD. Thus we achieve robustness under dif-

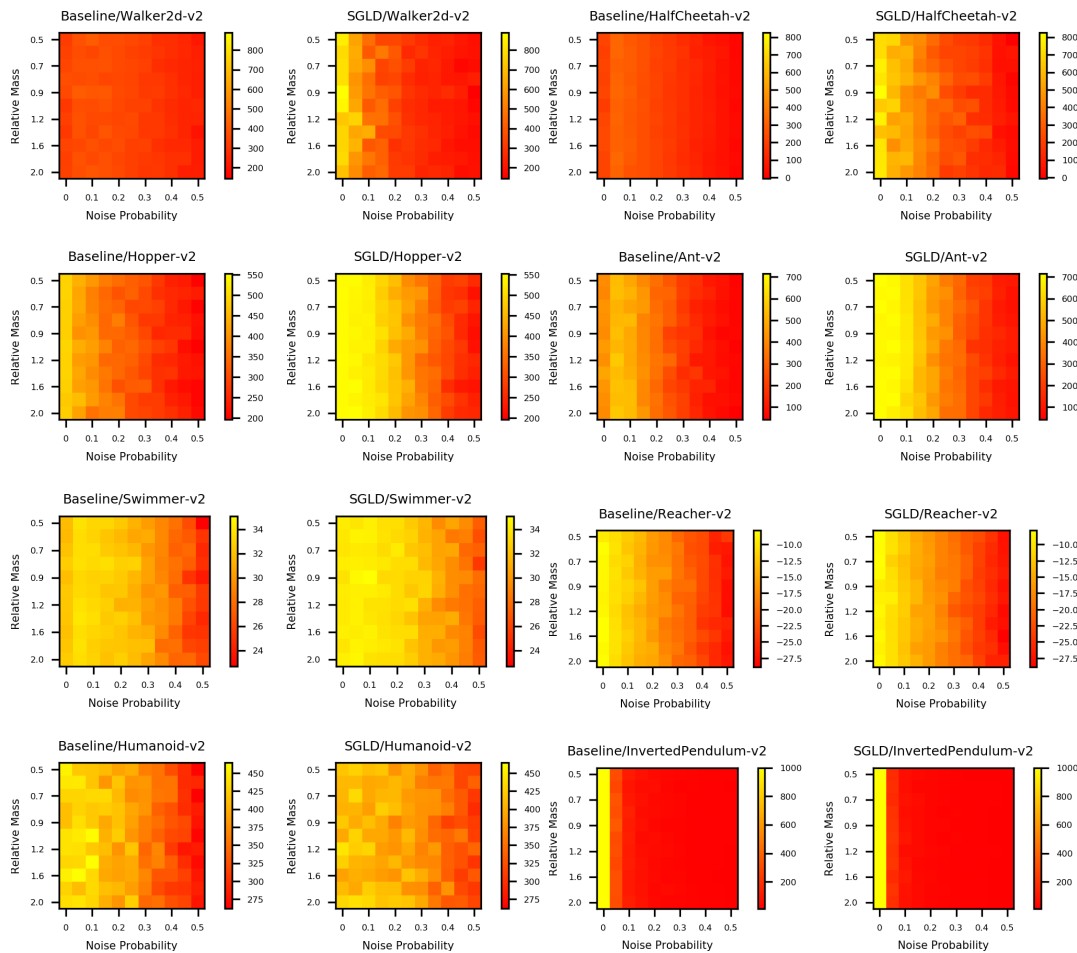

Figure 4: Average performance (over 5 seeds) of Algorithm 3 (—), and Algorithm 4 (—), under the NR-MDP setting with $\delta = 0$. The evaluation is performed on a range of noise probability and mass values not encountered during training.

ferent testing conditions with just a simple change in the DDPG algorithm and without additional computational cost.

## 5 CONCLUSION

In this work, we studied the robust reinforcement learning problem. By adapting the approximate infinite-dimensional entropic mirror descent from (Hsieh et al., 2019), we design a robust variant of DDPG algorithm, under the NR-MDP setting. In our experiments, we evaluated the robustness of our algorithm on several continuous control tasks, and found that our algorithm clearly outperformed the baseline algorithm from (Tessler et al., 2019).

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

## A  ALGORITHMS AND HYPERPARAMETER DETAILS

Most of the values for the hyperparameters (in Table 1) are chosen from Dhariwal et al. (2017). In addition, for Algorithm 3, we set

1. thermal noise $\sigma_t = \sigma_0 \times (1 - 5 \times 10^{-5})^t$, where $\sigma_0 \in \left\{10^{-2}, 10^{-3}, 10^{-4}, 10^{-5}\right\}$.
2. warmup steps $K_t = \min\left\{15, \lfloor(1 + 10^{-5})^t\rfloor\right\}$

The best performing values for every environment are presented in Tables 2 and 3.

Table 1: Table of Hyperparameters

| Hyperparameter | Value |
|---|---|
| critic optimizer | Adam |
| critic learning rate | $10^{-3}$ |
| target update rate $\tau$ | 0.999 |
| mini-batch size $N$ | 128 |
| discount factor $\gamma$ | 0.99 |
| damping factor $\beta$ | 0.9 |
| replay buffer size | $10^6$ |
| action noise parameter $\sigma$ | $\{0, 0.01, 0.1, 0.2, 0.3, 0.4\}$ |
| RMSProp parameter $\alpha$ | 0.999 |
| RMSProp parameter $\epsilon$ | $10^{-8}$ |
| RMSProp parameter $\eta$ | $10^{-4}$ |

Table 2: Hyperparameters chosen via grid search (for NR-MDP setting with $\delta = 0.1$)

| | Algorithm 3: $(\sigma_0, \sigma)$ | Algorithm 4: $\sigma$ |
|---|---|---|
| Walker-v2 | $(10^{-2}, 0.01)$ | 0 |
| HalfCheetah-v2 | $(10^{-2}, 0)$ | 0.01 |
| Hopper-v2 | $(10^{-3}, 0.2)$ | 0.2 |
| Ant-v2 | $(10^{-4}, 0.2)$ | 0.4 |
| Swimmer-v2 | $(10^{-5}, 0.4)$ | 0.4 |
| Reacher-v2 | $(10^{-3}, 0.2)$ | 0.4 |
| Humanoid-v2 | $(10^{-4}, 0.01)$ | 0 |
| InvertedPendulum-v2 | $(10^{-3}, 0.01)$ | 0.1 |

Table 3: Hyperparameters chosen via grid search (for NR-MDP setting with $\delta = 0$)

| | Algorithm 3: $(\sigma_0, \sigma)$ | Algorithm 4: $\sigma$ |
|---|---|---|
| Walker-v2 | $(10^{-2}, 0.1)$ | 0.01 |
| HalfCheetah-v2 | $(10^{-2}, 0.01)$ | 0.2 |
| Hopper-v2 | $(10^{-5}, 0.3)$ | 0.4 |
| Ant-v2 | $(10^{-2}, 0.4)$ | 0.4 |
| Swimmer-v2 | $(10^{-2}, 0.2)$ | 0.3 |
| Reacher-v2 | $(10^{-3}, 0.2)$ | 0.3 |
| Humanoid-v2 | $(10^{-2}, 0.1)$ | 0 |
| InvertedPendulum-v2 | $(10^{-3}, 0)$ | 0.01 |

## B  TOY EXAMPLE

**Setting.** We consider the following simplified/modified hide and seek (running and chasing) game from Baker et al. (2019):

- Two agents (a hider and a seeker) are placed in 2d-square environment without any obstacles. The hider wants to be away from the seeker as much as possible.
- The state (at time $t$) is represented by $s_t = \left(\ell_t^{\text{hide}}, \ell_t^{\text{seek}}, v_t^{\text{hide}}, v_t^{\text{seek}}\right) \in \mathcal{S}$, where $\ell_t^{\text{hide}}$ is the location of the hider, and $v_t^{\text{seek}}$ is the velocity of the seeker. We assume that both agents observe the full state.
- The action space is same for both agents, i.e., $a_t^{\text{hide}}, a_t^{\text{seek}} \in \mathcal{A} = \{\text{left}, \text{right}, \text{up}, \text{down}\}$.
- The reward function is given by $R_t = -R^{\text{hide}}(s_t) = R^{\text{seek}}(s_t) = \frac{1}{d\left(\ell_t^{\text{hide}}, \ell_t^{\text{seek}}\right)}$, i.e., inverse of the current distance between the hider and seeker.
- The additional parameters $f^{\text{hide}}$, and $f^{\text{seek}}$ denote the force of the hider and the seeker respectively. We fix $f^{\text{hide}} = 4$, and let $f^{\text{seek}} \in \{2, 3, 4\}$.
- The transition dynamics $T\left(s_{t+1} \mid s_t, a_t^{\text{hide}}, a_t^{\text{seek}}; f^{\text{hide}}, f^{\text{seek}}\right)$ depends on the parameters $f^{\text{hide}}$, and $f^{\text{seek}}$.
- We have pre-trained (three) competitive seeker policies $\bar{\pi}_{f^{\text{seek}}}$ for all $f^{\text{seek}} \in \{2, 3, 4\}$.

**Training.** For training, we fix setting with $f^{\text{hide}} = 4$, and $f^{\text{seek}} = 4$. We train the parametric policies $\pi_\theta^{\text{hide}}$, and $\pi_w^{\text{seek}}$ using the following two approaches (generate an episode $\ldots, S_t, A_t^{\text{hide}}, A_t^{\text{seek}}, R_t, S_{t+1}, \ldots$ using current parameters, and define $G_t = \sum_{\tau=0}^{T} \gamma^\tau R_{t+\tau}$):

- Gradient descent ascent (GDA) approach:

$$\theta_{t+1} \leftarrow \theta_t - \eta_t G_t \left[\nabla_\theta \ln \pi_\theta^{\text{hide}}\left(A_t^{\text{hide}} \mid S_t\right)\right]_{\theta=\theta_t}$$
$$w_{t+1} \leftarrow w_t + \eta_t G_t \left[\nabla_w \ln \pi_w^{\text{seek}}\left(A_t^{\text{seek}} \mid S_t\right)\right]_{w=w_t}$$

- Stochastic Gradient Langevin Dynamics (SGLD) approach:

$$\theta_{t+1} \leftarrow \theta_t - \eta_t G_t \left[\nabla_\theta \ln \pi_\theta^{\text{hide}}\left(A_t^{\text{hide}} \mid S_t\right)\right]_{\theta=\theta_t} + \sqrt{2\eta_t}\mathcal{N}\left(0, I\right)$$
$$w_{t+1} \leftarrow w_t + \eta_t G_t \left[\nabla_w \ln \pi_w^{\text{seek}}\left(A_t^{\text{seek}} \mid S_t\right)\right]_{w=w_t} + \sqrt{2\eta_t}\mathcal{N}\left(0, I\right)$$

We denote the final hider policy parameters resulting from the above methods as $\theta_{\text{GDA}}$, and $\theta_{\text{SGLD}}$.

**Evaluation.** We compare the performance of the policies $\pi_{\theta_{\text{GDA}}}^{\text{hide}}$, and $\pi_{\theta_{\text{SGLD}}}^{\text{hide}}$ under three different settings: $T\left(s_{t+1} \mid s_t, a_t^{\text{hide}}, a_t^{\text{seek}}; f^{\text{hide}}, f^{\text{seek}}\right)$, $a_t^{\text{seek}} \sim \bar{\pi}_{v^{\text{seek}}}\left(\cdot \mid s_t\right)$, $f^{\text{hide}} = 4$, and $f^{\text{seek}} \in \{2, 3, 4\}$. The results (averaged over 10 initial seeds, and 100 episodes per seed) are shown in Figure 5. In this simplified setting, on average, $\pi_{\theta_{\text{SGLD}}}^{\text{hide}}$ performs better than $\pi_{\theta_{\text{GDA}}}^{\text{hide}}$, but the difference in performance is not that significant as in the MuJoCo environments.

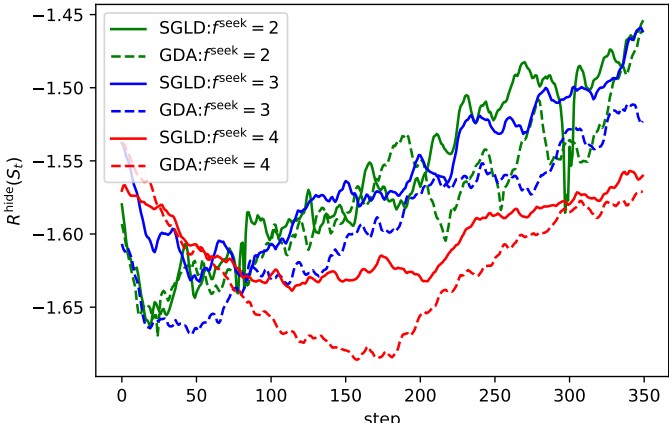

Figure 5: The instantaneous reward obtained by executing $\pi_{\theta_{\mathrm{GDA}}}^{\mathrm{hide}}$, and $\pi_{\theta_{\mathrm{SGLD}}}^{\mathrm{hide}}$ in different settings.

## C  ADDITIONAL RESULTS

The results for the best performing seeds are presented in Figures 6 and 7.

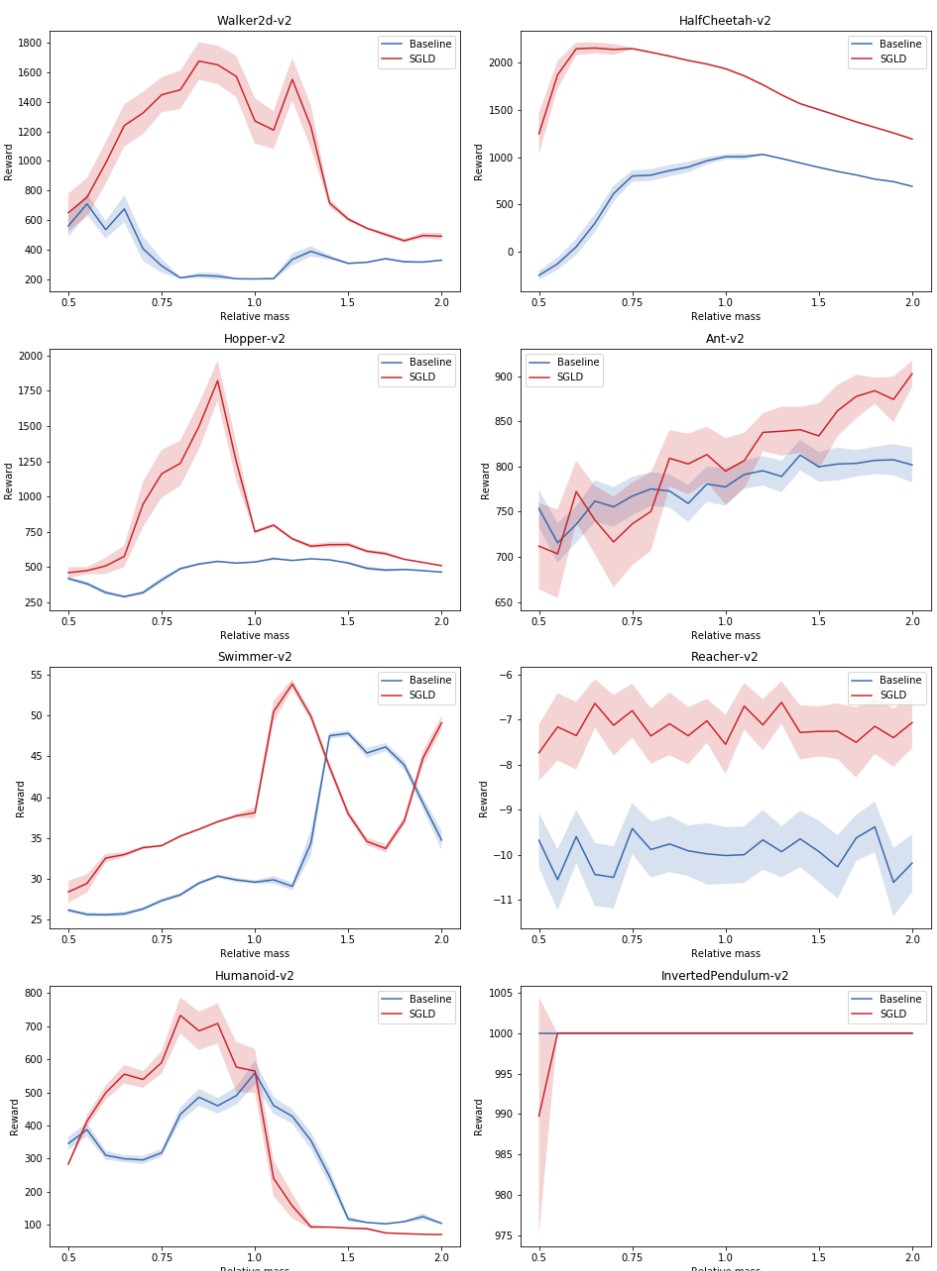

Figure 6: Average performance (of the best performing seed) of Algorithm 3 (—), and Algorithm 4 (—), under the NR-MDP setting with $\delta = 0.1$. The evaluation is performed without adversarial perturbations, on a range of mass values not encountered during training.

---

**Algorithm 3** Two-Player DDPG with SGLD Actors (SGLD)

---

**Hyperparameters:** see Table 1

Initialize (randomly) policy parameters $\omega_1, \theta_1$, and Q-function parameter $\phi$.

Initialize the target network parameters $\omega_{\text{targ}} \leftarrow \omega_1, \theta_{\text{targ}} \leftarrow \theta_1$, and $\phi_{\text{targ}} \leftarrow \phi$.

Initialize replay buffer $\mathcal{D}$.

Initialize $m \leftarrow \mathbf{0} \; ; \; m' \leftarrow \mathbf{0}$.

$t \leftarrow 1$.

**repeat**

  Observe state $s$, and select actions $a = \mu_{\theta_t}(s) + \xi \; ; \; a' = \nu_{\omega_t}(s) + \xi'$, where $\xi, \xi' \sim \mathcal{N}(0, \sigma I)$

  Execute the action $\bar{a} = (1 - \delta)a + \delta a'$ in the environment.

  Observe reward $r$, next state $s'$, and done signal $d$ to indicate whether $s'$ is terminal.

  Store $(s, \bar{a}, r, s', d)$ in replay buffer $\mathcal{D}$.

  If $s'$ is terminal, reset the environment state.

  **if** it's time to update **then**

    **for** however many updates **do**

      $\bar{\omega}_t, \omega_t^{(1)} \leftarrow \omega_t \; ; \; \bar{\theta}_t, \theta_t^{(1)} \leftarrow \theta_t$

      **for** $k = 1, 2, \ldots, K_t$ **do**

        Sample a random minibatch of $N$ transitions $B = \{(s, \bar{a}, r, s', d)\}$ from $\mathcal{D}$.

        Compute targets $y(r, s', d) = r + \gamma(1 - d) Q_{\phi_{\text{targ}}}\left(s', (1 - \delta)\mu_{\theta_{\text{targ}}}(s') + \delta\nu_{\omega_{\text{targ}}}(s')\right)$.

        Update critic by one step of (preconditioned) gradient descent using $\nabla_\phi L(\phi)$, where

$$L(\phi) \;=\; \frac{1}{N} \sum_{(s, \bar{a}, r, s', d) \in B} \left(y(r, s', d) - Q_\phi(s, \bar{a})\right)^2.$$

        Compute the (agent and adversary) policy gradient estimates:

$$\nabla_\theta \widehat{J(\theta, \omega_t)} \;=\; \frac{1 - \delta}{N} \sum_{s \in \mathcal{D}} \nabla_\theta \mu_\theta(s) \nabla_{\bar{a}} Q_\phi(s, \bar{a}) \big|_{\bar{a} = (1 - \delta)\mu_\theta(s) + \delta\nu_{\omega_t}(s)}$$

$$\nabla_\omega \widehat{J(\theta_t, \omega)} \;=\; \frac{\delta}{N} \sum_{s \in \mathcal{D}} \nabla_\omega \nu_\omega(s) \nabla_{\bar{a}} Q_\phi(s, \bar{a}) \big|_{\bar{a} = (1 - \delta)\mu_{\theta_t}(s) + \delta\nu_\omega(s)}.$$

        $g \leftarrow \left[\nabla_\theta \widehat{J(\theta, \omega_t)}\right]_{\theta = \theta_t^{(k)}} \; ; \; m \leftarrow \alpha m + (1 - \alpha) g \odot g \; ; \; C \leftarrow \text{diag}\left(\sqrt{m + \epsilon}\right)$

        $\theta_t^{(k+1)} \leftarrow \theta_t^{(k)} + \eta C^{-1} g + \sqrt{2\eta}\sigma_t C^{-\frac{1}{2}}\xi$, where $\xi \sim \mathcal{N}(0, I)$

        $g' \leftarrow \left[\nabla_\omega \widehat{J(\theta_t, \omega)}\right]_{\omega = \omega_t^{(k)}} \; ; \; m' \leftarrow \alpha m' + (1 - \alpha) g' \odot g' \; ; \; D \leftarrow \text{diag}\left(\sqrt{m' + \epsilon}\right)$

        $\omega_t^{(k+1)} \leftarrow \omega_t^{(k)} - \eta D^{-1} g' + \sqrt{2\eta}\sigma_t D^{-\frac{1}{2}}\xi'$, where $\xi' \sim \mathcal{N}(0, I)$

        $\bar{\omega}_t \leftarrow (1 - \beta)\bar{\omega}_t + \beta\omega_t^{(k+1)} \; ; \; \bar{\theta}_t \leftarrow (1 - \beta)\bar{\theta}_t + \beta\theta_t^{(k+1)}$

        Update the target networks:

$$\phi_{\text{targ}} \;\leftarrow\; \tau\phi_{\text{targ}} + (1 - \tau)\phi$$

$$\theta_{\text{targ}} \;\leftarrow\; \tau\theta_{\text{targ}} + (1 - \tau)\theta_t^{(k+1)}$$

$$\omega_{\text{targ}} \;\leftarrow\; \tau\omega_{\text{targ}} + (1 - \tau)\omega_t^{(k+1)}$$

      **end for**

      $\omega_{t+1} \leftarrow (1 - \beta)\omega_t + \beta\bar{\omega}_t \; ; \; \theta_{t+1} \leftarrow (1 - \beta)\theta_t + \beta\bar{\theta}_t$

      $t \leftarrow t + 1$.

    **end for**

  **end if**

**until** convergence

**Output:** $\omega_T, \theta_T$.

---

---

**Algorithm 4** Two-Player DDPG with RMSProp Actors (Baseline)

---

**Hyperparameters:** see Table 1
Initialize (randomly) policy parameters $\omega_1, \theta_1$, and Q-function parameter $\phi$.
Initialize the target network parameters $\omega_{\text{targ}} \leftarrow \omega_1$, $\theta_{\text{targ}} \leftarrow \theta_1$, and $\phi_{\text{targ}} \leftarrow \phi$.
Initialize replay buffer $\mathcal{D}$.
Initialize $m \leftarrow \mathbf{0}$ ; $m' \leftarrow \mathbf{0}$.
$t \leftarrow 1$.
**repeat**
    Observe state $s$, and select actions $a = \mu_{\theta_t}(s) + \xi$ ; $a' = \nu_{\omega_t}(s) + \xi'$, where $\xi, \xi' \sim \mathcal{N}(0, \sigma I)$
    Execute the action $\bar{a} = (1 - \delta)a + \delta a'$ in the environment.
    Observe reward $r$, next state $s'$, and done signal $d$ to indicate whether $s'$ is terminal.
    Store $(s, \bar{a}, r, s', d)$ in replay buffer $\mathcal{D}$.
    If $s'$ is terminal, reset the environment state.
    **if** it's time to update **then**
        **for** however many updates **do**
            Sample a random minibatch of $N$ transitions $B = \{(s, \bar{a}, r, s', d)\}$ from $\mathcal{D}$.
            Compute targets $y(r, s', d) = r + \gamma(1 - d) Q_{\phi_{\text{targ}}}\left(s', (1 - \delta)\mu_{\theta_{\text{targ}}}(s') + \delta\nu_{\omega_{\text{targ}}}(s')\right)$.

            Update critic by one step of (preconditioned) gradient descent using $\nabla_\phi L(\phi)$, where

$$L(\phi) = \frac{1}{N} \sum_{(s, \bar{a}, r, s', d) \in B} (y(r, s', d) - Q_\phi(s, \bar{a}))^2.$$

            Compute the (agent and adversary) policy gradient estimates:

$$\nabla_\theta \widehat{J(\theta, \omega_t)} = \frac{1 - \delta}{N} \sum_{s \in \mathcal{D}} \nabla_\theta \mu_\theta(s) \nabla_{\bar{a}} Q_\phi(s, \bar{a})\big|_{\bar{a} = (1 - \delta)\mu_\theta(s) + \delta\nu_{\omega_t}(s)}$$

$$\nabla_\omega \widehat{J(\theta_t, \omega)} = \frac{\delta}{N} \sum_{s \in \mathcal{D}} \nabla_\omega \nu_\omega(s) \nabla_{\bar{a}} Q_\phi(s, \bar{a})\big|_{\bar{a} = (1 - \delta)\mu_{\theta_t}(s) + \delta\nu_\omega(s)}.$$

            $g \leftarrow \left[\nabla_\theta \widehat{J(\theta, \omega_t)}\right]_{\theta = \theta_t}$ ; $m \leftarrow \alpha m + (1 - \alpha) g \odot g$ ; $C \leftarrow \text{diag}\left(\sqrt{m + \epsilon}\right)$
            $\theta_{t+1} \leftarrow \theta_t + \eta C^{-1} g$
            $g' \leftarrow \left[\nabla_\omega \widehat{J(\theta_t, \omega)}\right]_{\omega = \omega_t}$ ; $m' \leftarrow \alpha m' + (1 - \alpha) g' \odot g'$ ; $D \leftarrow \text{diag}\left(\sqrt{m' + \epsilon}\right)$
            $\omega_{t+1} \leftarrow \omega_t - \eta D^{-1} g'$
            Update the target networks:

$$\phi_{\text{targ}} \leftarrow \tau\phi_{\text{targ}} + (1 - \tau)\phi$$
$$\theta_{\text{targ}} \leftarrow \tau\theta_{\text{targ}} + (1 - \tau)\theta_{t+1}$$
$$\omega_{\text{targ}} \leftarrow \tau\omega_{\text{targ}} + (1 - \tau)\omega_{t+1}$$

            $t \leftarrow t + 1$.
        **end for**
    **end if**
**until** convergence
**Output:** $\omega_T, \theta_T$.

---

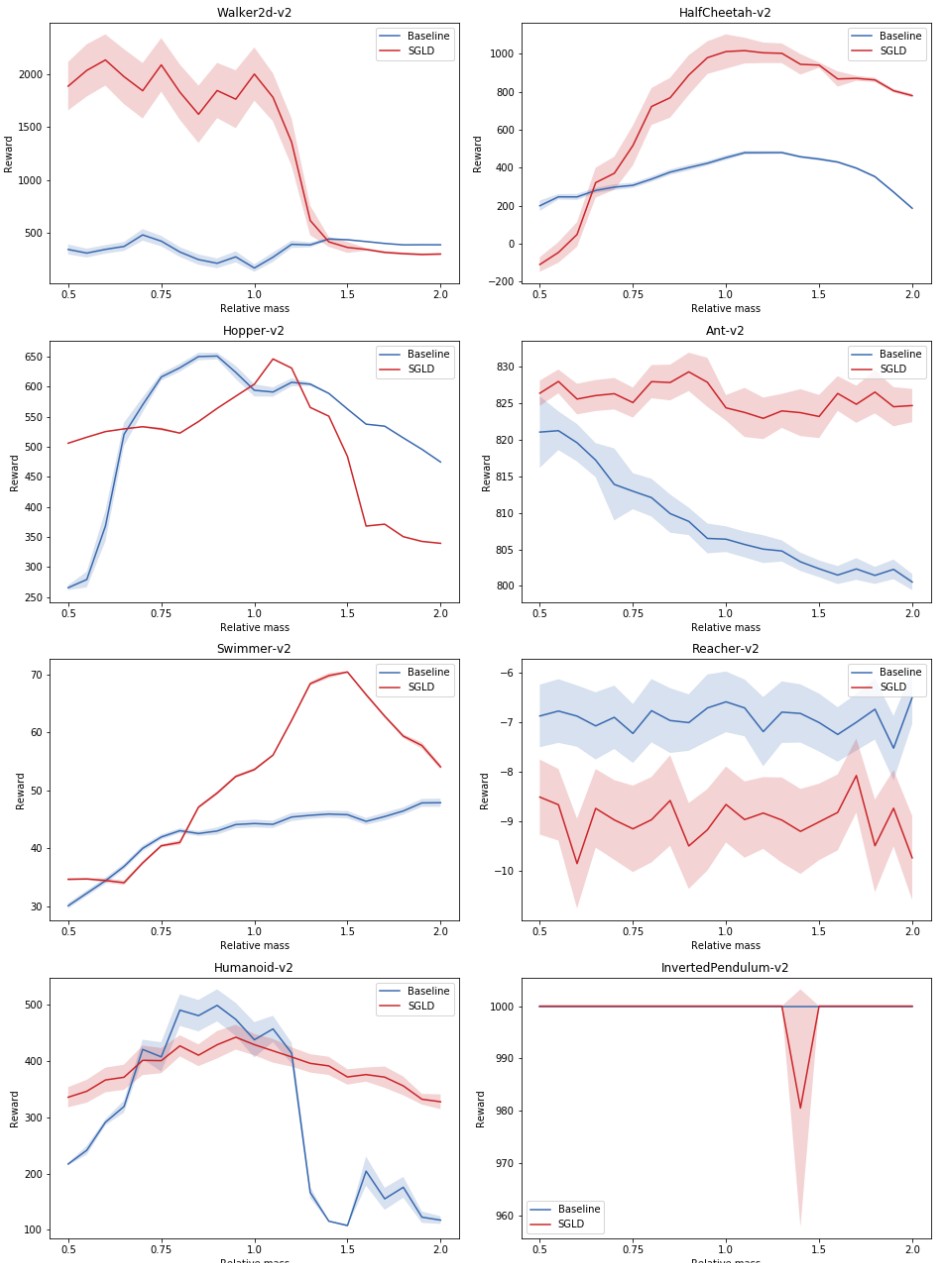

Figure 7: Average performance (of the best performing seed) of Algorithm 3 (——), and Algorithm 4 (——), under the NR-MDP setting with $\delta = 0$. The evaluation is performed without adversarial perturbations, on a range of mass values not encountered during training.

