# OpenReview forum: "Robust Reinforcement Learning via Adversarial Training with  Langevin Dynamics"
_ICLR.cc/2020/Conference — Reject_

### Official Review · AnonReviewer1 · 2019-10-20
**Official Blind Review #1**

**Rating:** 3

**Review:**

Update: I have read the author responses. As mentioned before I don't have the background to carefully assess the experiments and will have to rely on my fellow reviewers here. However I stand by my opinion that the experimental results would need to be very strong to warrant an acceptance, since the conceptual contribution is relatively limited.
I was also disappointed by the authors unwillingness to back up their claims of "theoretically convergent" with a proof, or at least a theorem. Therefore I still tend towards rejecting the paper.
-----------------------------------------------
Summary
The present work proposes to use the recently developed Stochastic Gradient Langevin Dynamics (SGLD) to compute mixed approximate equilibria in two-player reinforcement learning, following the methodology proposed by hsieh et al for generative adversarial networks. The authors report practical improvements compared to pure strategies computed as proposed by Tessler et al.

Decision
The idea of using randomized strategies for two-player reinforcement learning is interesting and natural. However, as the authors note, mixed strategies are classical in game theory. Furthermore, the adaption of the methodology of hsieh et al is straightforward, limiting the strength of the theoretical contribution.
Unfortunately, the theoretical contribution is not stated consistently, since in the introduction, the paper states "Our paper precisely bridges this gap between theory and practice in previous works, by proposing the ﬁrst theoretically convergent algorithm for robust RL", but this claim is missing in the abstract or conclusion, and there is no theorem that justifies this rather strong claim.
In my opinion, this paper should only be accepted if it provides very convincing numerical experiments, which I am not qualified to assess. I happy to increase my score if the experimental results are deemed strong by the reviewers with more expertise in practical reinforcement learning.

Questions to authors
You write "Our paper precisely bridges this gap between theory and practice in previous works, by proposing the ﬁrst theoretically convergent algorithm for robust RL". What is the exact mathematical statement here? Does this refer to the algorithm that is used in the numerical experiments?

**Experience Assessment:**

I do not know much about this area.

**Review Assessment: Checking Correctness Of Derivations And Theory:**

N/A

**Review Assessment: Checking Correctness Of Experiments:**

I assessed the sensibility of the experiments.

**Review Assessment: Thoroughness In Paper Reading:**

I made a quick assessment of this paper.

---

### Official Review · AnonReviewer2 · 2019-10-23
**Official Blind Review #2**

**Rating:** 3

**Review:**

This paper approaches two-player zero-sum Markov game by using mixed Nash equilibrium by sampling from randomized policies. The authors propose that the optimization is done using Stochastic Gradient Langevin Dynamics iterations.
In my opinion, the use of SGLD to this problem is potentially useful - as partially proved by this work. However, this application is obvious and does not have enough novelty merits to be accepted to this ICLR. The experiment make senses but is not rigorous enough to assure the practitioners on the improvement over baseline.
I recommend the authors to further develop this idea in both theoretical improvement and experiment settings to further explore the direction.

**Experience Assessment:**

I do not know much about this area.

**Review Assessment: Checking Correctness Of Derivations And Theory:**

I assessed the sensibility of the derivations and theory.

**Review Assessment: Checking Correctness Of Experiments:**

I assessed the sensibility of the experiments.

**Review Assessment: Thoroughness In Paper Reading:**

I made a quick assessment of this paper.

---

### Official Review · AnonReviewer3 · 2019-10-23
**Official Blind Review #3**

**Rating:** 6

**Review:**

The paper proposes a new algorithm for solving Noisy Robust MDPs (NR-MDPs) based on computing an approximate gradient using  stochastic gradient langevin dynamics (SGLD). THE NR-MDPs can be thought of as an agent learning in an adversarial environment.
The paper is well written and the references pointed to give sufficient background to understand the problem undertaken.
Most of the theory comes from the "Finding mixed nash equilibria of generative adversarial networks." (ICML, 2019), and the papers main contribution lies in applying the theory to reinforcement learning.
It is not altogether unexpected though that Langevin Dynamics give better results than the more standard gradient-based approach considered for saddle-point problems, that's what they are supposed to do. But, this seems to be the first time SGLD has been applied to such a problem.
The authors compare on the MuJoCo benchmark with common but not identical instances to "Action robust reinforcement learning and applications in continuous control", which also provides the baseline algorithm used for comparison. Mentioned in this paper is the difficulty of solving the "inverted pendulum" instance by the algorithm proposed therein. It is infact mentioned as a failure case. Maybe, the authors can show results on the same.
Additionally, maybe it would make sense to have an ablation study of the hyper-parameters from Table 1.


**Experience Assessment:**

I have read many papers in this area.

**Review Assessment: Checking Correctness Of Derivations And Theory:**

I assessed the sensibility of the derivations and theory.

**Review Assessment: Checking Correctness Of Experiments:**

I assessed the sensibility of the experiments.

**Review Assessment: Thoroughness In Paper Reading:**

I read the paper at least twice and used my best judgement in assessing the paper.

---

### Official Review · AnonReviewer4 · 2019-11-11
**Official Blind Review #4**

**Rating:** 3

**Review:**

Summary:
The paper considers the task of learning robust policies. Specifically, they focus on the noisy-robust (NR) variant of the action robust framework proposed in [1]. As noted in [1], as strong duality does not hold in the NR variant it can not be solved using deterministic policies (pure strategies).
The authors propose to combine SGLD with DDPG, to maintain a distribution over deterministic policies (essentially a mixed strategy) - hence overcoming these issues.
The results look promising, outperforming [1] in all tested domains.

Review:
Overall this seems like the first approach of using Bayesian learning in order to reach mixed Nash equilibrium in RL. This is super-important, since many problems can be formulated as zero-sum games for which the solution is not necessarily a pure strategy.
As [2] have already shown the ability of this concept to find mixed equilibria in GANs, the main novelty is the introduction to RL.

As this work does not introduce new theory or a dramatic new concept, I feel that the acceptance of this work lies mainly on the empirical side. In my opinion, the experiments need to be more convincing - for instance, include additional domains (such as the Inverted Pendulum which was a failure case in [1]) and additional forms of robustness (e.g., the probabilistic-robust variant from [1] which is based on deterministic strategies and was shown to work better, and RARL [3] which test robustness to external disturbances).
An addition option is to build a low-dimensional toy problem in which the exact solution is known. This will enable you to show that while the naive solution in [1] either does not converge or converges to sub-optimal solutions, the SGLD approach is capable of finding superior solutions (hopefully the global optimum).

The idea is in the right direction. However, in my opinion, it is not there yet and is thus not ready for ICLR.

--- Post Rebuttal ---

I stand by my original assessment. I feel that such work needs to be more convincing. I for one would feel more confident had the authors provided simpler experiments in which they show that their approach indeed converges to the mixed Nash equilibria while the NR-DDPG approach from [1] does not.
I did overall like this direction and I believe Robustness in RL is very important.


[1] Action Robust Reinforcement Learning and Applications in Continuous Control - Tessler et al. 2019
[2] Finding mixed nash equilibria of generative adversarial networks - Hsieh et al. 2019
[3] Robust Adversarial Reinforcement Learning - Pinto et al. 2017

**Experience Assessment:**

I have published in this field for several years.

**Review Assessment: Checking Correctness Of Derivations And Theory:**

I assessed the sensibility of the derivations and theory.

**Review Assessment: Checking Correctness Of Experiments:**

I carefully checked the experiments.

**Review Assessment: Thoroughness In Paper Reading:**

I read the paper thoroughly.

---

### Author Response · Authors · 2019-11-14
**on the strength of the numerical evidence -- inverted pendulum works perfectly**

Dear Reviewers,

We now include new numerical evidence to support our case:

1. We obtain superior performance on the inverted pendulum, which is the failure case for [1]; cf., Figures 1 and 3.

Regarding these two figures, we would like to emphasize that the baseline in Figure 1 is the action robust-DDPG algorithm proposed in [1], and the baseline in Figure 3 is the standard non-robust DDPG algorithm, which is the baseline for [1].

2. We include additional results with MuJoCo environments: Swimmer, Reacher, and Humanoid. SGLD-DDPG performs clearly better than the baseline [1] in these examples.

Regarding the additional as well as the previous results, note that we have done all the standard/common set of experiments reported in robust deep-RL papers. In particular, please compare our numerical evidence and that of [1]: Both have 8 MuJoCo environments. Finally, similar to [1], we have also followed the best practices prescribed in [2].

3. We are currently trying to setup the hide and seek environment from openAI, which creates a stylized two-player case. Since the setting is simple, we expect both our algorithm and the baseline to perform similarly; however, it is nonetheless a great warm up exercise to show the contrast that follows with superior performance on the MuJoCo environment as well as the inverted pendulum. We hope to include this result before the rebuttal deadline; we can for sure include it in the camera ready, if accepted.

In summary, we believe that our numerical evidence is more than sufficient. In the light of this, we respectfully ask the reviewers to reconsider our score.

best,
Authors



[1] Action Robust Reinforcement Learning and Applications in Continuous Control - Tessler et al. 2019

---

### Author Response · Authors · 2019-11-14
**numerical evidence is strong and our *technical contribution* is also sufficient.**

Dear Reviewers,

The reviewers criticize that the theoretical contribution may not be sufficient and that the numerical evidence should be stronger.

We included additional, stronger numerical evidence in the new revision, including capturing a failure case in the baseline. Please see further our previous comment.

We now would like to argue that the technical contribution is also sufficient.

First of all, thanks for acknowledging that our approach as the first approach of using SGLD to reach the mixed Nash equilibrium in reinforcement learning (RL). The mixed Nash perspective should be investigated further as it also sheds light into popular problems, such as self-play. As a result, bringing this to the attention of the literature is important.

While the two player algorithm via the Langevin dynamics is the same algorithm from [1], there is no need to improve on it since it is already theoretically grounded. We do not need to provide new infinite dimensional convergence theorems to show that it is useful in RL. We thought that the mixed setting itself is novel and that we do not need to obfuscate the contributions, and hence, we relied on a state-of-the-art algorithm [1].

However, we observe in our paper that the algorithm in [1] is computationally demanding even though it uses a mean approximation in its inner loop (which already saves significant resources).

As a result, we introduced a new approximation by setting delta=0 and K_t=1. You can see this as just applying entropic mirror descent with a short inner loop for a sampling problem that has a dirac delta solution (i.e., there is no mixed solution but just a pure solution).

Even in this pure case, the new algorithm performs superior to the baseline (i.e., non-robust DDPG algorithm) with similar per iteration computational complexity to DDPG itself; cf., Figure 3. This simultaneous improvement should also not be understated since it has implications in all the applications of DDPG.

In the light of these observations, we respectfully ask the reviewers to acknowledge that our novelty is sufficient for publication.

best,
Authors


[1] Finding mixed nash equilibria of generative adversarial networks - Hsieh et al. 2019

---

> ### Comment · AnonReviewer1 · 2019-11-14
> **repeated question**
>
> Questions to authors
> You write "Our paper precisely bridges this gap between theory and practice in previous works, by proposing the ﬁrst theoretically convergent algorithm for robust RL". What is the exact mathematical statement here? Does this refer to the algorithm that is used in the numerical experiments?

---

> > ### Comment · AnonReviewer4 · 2019-11-14
> > **Additional remark**
> >
> > I am almost certain that in [1] (the baseline) the authors provide two forms of robustness, named PR and NR. While the NR has problems, as the authors have stated in the paper, the PR does have convergence guarantees in RL (in the tabular case).

---

> > > ### Author Response · Authors · 2019-11-15
> > > **additional remark**
> > >
> > > We will clearly acknowledge that the PR-MDP setting in [2] has convergence guarantees in the text when we update to add the openAI hide-and-seek environment benchmark.
> > >
> > > Note however that our work considers the problematic NR-MDP setting.

---

> > ### Author Response · Authors · 2019-11-15
> > **answer to the question**
> >
> > Any general min-max (mixed-NE) problem given in Eq.(3) can be solved using the infinite-dimensional mirror descent algorithm (Algorithm 2 in the appendix), with provable convergence guarantee [1]. In section 2.2, we briefly reviewed the existing theory from [1]. In section 3, we translated the two-player RL problem in the mixed-NE form, c.f., Eq.(5). Then the theory from [1] naturally follows. For experiments, we used Algorithm 1, adapted to DDPG.
> >
> > We can further clarify the remark in the paper. We can also tone this down, if reviewers find it too ambitious.
> >
> > [1] Finding mixed nash equilibria of generative adversarial networks - Hsieh et al. 2019

---

### Decision · Program_Chairs · 2019-12-19

**Decision:**

Reject

**Comment:**

The authors address the problem of robust reinforcement learning. They propose an adversarial perspective on robustness. Improving the robustness can now be seen as two agent playing a competitive game, which means that in many cases the first agent needs to play a mixed strategy. The authors propose an algorithm for optimizing such mixed strategies.

Although the reviewers are convinced of the relevance of the work (as a first approach of Bayesian learning to reach mixed Nash equilibria, which is useful not only for robustness but for any problem that can be formulated as zero-sum game requiring a mixed strategy), they are not completely convinced by the work in current state. Three of the reviewers commented on the experiments not being rigorous and convincing enough in current form, and thus not (yet!) being able to recommend acceptance to ICLR.